# Uncovering Emotional and Identity-Driven Dimensions of Entertainment Consumption in a Transitional Digital Culture

**DOI:** 10.3390/bs15081049

**Published:** 2025-08-01

**Authors:** Ștefan Bulboacă, Gabriel Brătucu, Eliza Ciobanu, Ioana Bianca Chițu, Cristinel Petrișor Constantin, Radu Constantin Lixăndroiu

**Affiliations:** 1Department of Marketing, Tourism Services and International Affairs, Faculty of Economic Sciences and Business Administration, Transilvania University of Brașov, Colina Universității Street, No. 1, Building A, 500068 Brașov, Romania; stefan.bulboaca@unitbv.ro (Ș.B.); gabriel.bratucu@unitbv.ro (G.B.); ioana.chitu@unitbv.ro (I.B.C.); cristinel.constantin@unitbv.ro (C.P.C.); 2Department of Management and Economic Informatics, Faculty of Economic Sciences and Business Administration, Transilvania University of Brașov, Colina Universității Street, No. 1, Building A, 500068 Brașov, Romania; lixi.radu@unitbv.ro

**Keywords:** entertainment consumption, consumer behavior, qualitative analysis, inductive and deductive coding, emotional needs, identity construction, belonging, validation

## Abstract

This study explores entertainment consumption patterns in Romania, a transitional digital culture characterized by high digital connectivity but underdeveloped physical infrastructure. Employing a dual qualitative coding methodology, this research combines inductive analysis of consumer focus groups with deductive analysis of expert interviews, enabling a multi-layered interpretation of both overt behaviors and latent emotional drivers. Seven key thematic dimensions, motivational depth, perceived barriers, emotional needs, clarity of preferences, future behavioral intentions, social connection, and identity construction, were analyzed and compared using a Likert-based scoring framework, supported by a radar chart and comparison matrix. Findings reveal both convergence and divergence between consumer and expert perspectives. While consumers emphasize immediate experiences and logistical constraints, experts uncover deeper emotional motivators such as validation, mentorship, and identity formation. This behavioral–emotional gap suggests that, although digital entertainment dominates due to accessibility, it often lacks the emotional richness associated with physical formats, which are preferred but less accessible. This study underscores the importance of triangulated qualitative inquiry in revealing not only stated preferences but also unconscious psychological needs. It offers actionable insights for designing emotionally intelligent and culturally responsive entertainment strategies in digitally saturated yet infrastructure-limited environments.

## 1. Introduction

In today’s hyperconnected society, entertainment plays a critical role beyond leisure because it operates as a psychological space where individuals seek emotional grounding, construct identity, and form social bonds. Particularly for younger generations, entertainment has become a primary outlet for self-expression, emotional regulation, and social integration. As traditional community structures shift and cultural experiences become increasingly mediated through digital platforms, understanding how people engage with entertainment becomes essential for both scholars and practitioners aiming to navigate this evolving cultural terrain.

Digitalization has revolutionized access to entertainment, making it more ubiquitous, individualized, and on-demand. Yet this convenience often comes at the cost of emotional depth. Emerging evidence suggests that, while digital entertainment offers functional benefits, such as affordability, flexibility, and constant availability, it frequently fails to satisfy deeper psychological needs like belonging, validation, and identity affirmation. Conversely, physical entertainment experiences (festivals, concerts, live events) are perceived as more emotionally resonant but are often hindered by barriers such as cost, time, and infrastructure limitations. These tensions create a behavioral paradox: individuals gravitate toward digital formats out of necessity but continue to express a preference for the emotional richness of physical experiences.

This paradox is particularly pronounced in Romania, a country characterized by high digital connectivity but a fragmented and underdeveloped physical entertainment infrastructure. As a result, Romania exemplifies what this study refers to as a transitional digital culture, in which behavioral habits, technological access, and emotional expectations frequently misalign. Understanding entertainment consumption in such a context is critical to unpack how individuals negotiate between digital and physical formats and, also, to illuminate the unspoken emotional drivers that shape these decisions. Addressing this gap can offer actionable insights for marketers, policymakers, and cultural producers seeking to design entertainment experiences that are not only accessible but emotionally intelligent and socially responsive.

This study adopts a dual-method qualitative design to investigate both consumer and expert perspectives on entertainment consumption in Romania. While consumers offer insight into their lived experiences and articulated preferences, industry specialists provide an interpretive lens that reveals latent motivations and structural constraints. The research is guided by three questions:(1)How do specialists perceive consumer needs and preferences in the entertainment industry?(2)What are consumers’ expressed and unspoken motivations in engaging with entertainment?(3)How do these perspectives align or diverge, and what strategic implications emerge for market segmentation, positioning, and communication?

By integrating inductive and deductive analysis across these stakeholder groups, this study contributes to a deeper understanding of the emotional architecture of entertainment consumption in digitally evolving societies. It positions entertainment not merely as a cultural product but as a vital channel for identity construction and emotional fulfillment—domains increasingly central to consumer behavior in the 21st century.

This study reveals a profound divergence between the stated preferences and actual behaviors of Romanian entertainment consumers, highlighting the psychological and structural tensions in a transitional digital culture. While consumers express a clear preference for physical entertainment experiences, such as festivals and social gatherings, digital entertainment dominates their routines due to accessibility, affordability, and convenience. This behavior–preference mismatch underscores a deeper emotional disconnect such as physical formats are more fulfilling but less available, while digital formats are habitual yet emotionally unsatisfying.

By exploring these dynamics through both inductive and deductive analyses, this paper sheds light on the emotional architectures that shape entertainment consumption in transitional digital cultures. This study offers insights relevant to marketers, policymakers, and cultural producers seeking to design emotionally intelligent and socially responsive entertainment experiences.

## 2. Literature Review

Consumers typically engage with products and services to fulfill specific needs and desires with happiness emerging as a secondary outcome upon goal attainment. In contrast, entertainment consumption is characterized by a more immediate pursuit of emotional gratification. Consequently, over the past five centuries, prevailing definitions of entertainment have consistently emphasized its association with consumer interest, enjoyment, and satisfaction ([36]), including four main components: psychological, cognitive, affective, and behavioral ([10]).

Consumer preferences and transitional digital culture have shifted dramatically toward digital platforms, driven by convenience, affordability, and on-demand availability. This trend has restructured traditional entertainment consumption models ([53]). As digital entertainment platforms desired by convenience-seeking consumers emerge ([34]), the way they interact with entertainment is changing, resulting in an evolving entertainment ecosystem ([26]). This transition has transformed and redefined the experience offered by cultural industries “from superficial, homogeneous, and unidirectional to profound, diverse, and interactive” ([18]), increasing the “audience level of active engagement and co-creation” ([40]). Basaran and Ventura’s study reaches the same conclusion: digital entertainment companies need to be highly adaptable and dynamic. Due to the fact that consumer tastes and needs are changing rapidly, digital consumer engagement is one of the most important tools that can be used to face these challenges ([6]). However, specialists also consider other effects of digitalization related to the “creative destructive impact on non-digitized elements” ([17]), digital fatigue, or cognitive overload ([21]; [28]; [45]; [47]).

While digital entertainment offers accessibility and flexibility, it often lacks the emotional richness associated with physical formats like live events and festivals. Consumers report feeling more emotionally fulfilled through physical interactions. To effectively meet consumer needs, it is essential to understand their underlying needs and desires. Motivation theories, including Maslow’s hierarchy of needs, remain relevant in interpreting entertainment behaviors, particularly regarding the pursuit of psychological security, belonging, and self-actualization. These needs influence both digital and physical entertainment choices ([37]; [1]). Maslow’s hierarchy of needs offers a foundational framework in this regard, proposing five distinct levels of human motivation. According to his theory, the satisfaction of a current need facilitates the progression to higher-level needs, highlighting the dynamic and sequential nature of consumer motivation ([37]; [1]).

For example, streaming services like Netflix have transformed viewer behavior, promoting binge-watching and diminishing traditional viewing routines. These platforms foster habitual media engagement and influence user expectations for personalized content ([32]). Moreover, recommendation engines employing machine learning algorithms enhance digital entertainment by providing curated experiences based on user data and preferences. This personalization increases content discoverability and platform loyalty. Additionally, big data analytics has become essential in the entertainment industry, enabling content personalization, real-time audience behavior analysis, and revenue optimization. These tools allow providers to respond rapidly to consumer preferences and enhance user satisfaction ([2]).

The study by [5] ([5]) investigates the adoption of online theater streaming services in Hungary during the COVID-19 pandemic, utilizing the Unified Theory of Acceptance and Use of Technology 2 model. The research identifies “habit” as the most significant predictor of both behavioral intention and actual usage of theater webcasting, surpassing factors like hedonic motivation and price value. This suggests that the habits formed during lockdowns could underpin sustainable digital business models for theaters in the post-pandemic era. Additionally, the study finds no significant generational differences in adoption patterns, indicating a uniform response across age groups under the specific social and technological conditions imposed by the pandemic.

Belonging and identity construction are key emotional needs that drive entertainment consumption, particularly in transitional digital cultures. Entertainment serves as a space for social validation and personal expression ([3]; [48]). Also, the fear of missing out can shape impulsive digital entertainment behaviors, particularly among youth. Social influence, real-time updates, and the desire for social inclusion reinforce compulsive consumption patterns ([27]; [30]; [31]).

The need for belonging represents a key element in consumption behavior that is correlated with any social interaction. Belonging is the need to connect with other people positively based on affiliation motivation. Moreover, this need is one of the most representative needs that builds an individual’s identity. However, belonging is not dependent on proximity to other people but rather on the perception of the quality of social connections with other people ([3]).

In this sense, belonging reflects a person’s perception of their involvement in a social environment or system. Thus, this need plays a vital role in defining a person’s identity from the age of 10 and even up to 24 years because, during adolescence, an individual is very prone to avoid social risks to avoid being socially exposed ([48]). Experts in the entertainment sector recognize a growing need to create identity-affirming and emotionally intelligent experiences, particularly in digitally saturated but infrastructure-limited societies. The study by [33] ([33]) offers a comprehensive systematic review of consumer engagement in digital entertainment, analyzing 204 scholarly articles from five reputable publishers. After a rigorous screening process, 24 articles were deemed highly relevant and subjected to content analysis. The findings highlight a predominant focus on social media platforms, particularly Facebook, Twitter, and Instagram, as primary channels for enhancing consumer engagement in digital entertainment. Notably, the year 2015 marked the highest number of citations in this domain, with a significant increase in publications observed post-2017. Geographically, the United States led in research contributions, followed by Australia, the United Kingdom, China, and India. This review underscores the critical role of social media in shaping consumer engagement strategies within the digital entertainment industry and identifies areas for future research exploration. These strategies aim to bridge the gap between behavior and emotional fulfillment.

Validation is a feeling and desire of people that is present in the form of social acceptance, or at least the situation in which people around them do not have negative opinions towards their views. This need has direct implications, such as improving the performance and efficiency of an individual in specific situations, as well as easier acceptance of some tendencies (if there is at least one person in proximity who has already validated that tendency) ([52]).

Following the identification of the key aspects influencing consumer behavior, the next step involves managing the experience itself. In this context, the work of M. Yerkes and John D. Dodson is particularly relevant. Their research emphasizes the importance of identifying optimal levels of stimulus intensity (curiosity, excitement) for effective performance ([29]). According to the Yerkes–Dodson Law, there is a curvilinear relationship between arousal and performance, suggesting that moderate levels of arousal enhance engagement, while too little or too much can hinder an individual’s ability to successfully participate in or complete a task. Also, not only the implications for stress and memory ([23]) but also the influencing possibilities of the peak ([7]) are important aspects to take into account.

Moreover, Langer (1975) mentions that the level of information that a person has changes his perception of his own control over certain events, and the increase in known information can cause him to misjudge reality, a theory confirmed lately ([11]; [20]; [49]). Also, the possibility of initiation of consumption can be stimulated through the interenvironment of the group effect. In this sense, there is the “Fear of Missing Out” concept showing up, which defines an impulsive behavior of action or reaction with the aim of maintaining a social connection ([27]; [30]; [31]).

Building on existing frameworks, this study integrates Maslow’s hierarchy of needs with identity construction theory, particularly in transitional digital cultures. By emphasizing “transitional digital culture” as a bridging concept, we highlight how hybrid consumption behaviors (digital and physical) reflect evolving consumer identity dynamics specific to post-socialist Eastern European contexts.

## 3. Materials and Methods

This study employed qualitative research design combining focus groups with entertainment consumers and in-depth interviews with industry experts to explore both explicit behaviors and latent emotional drivers of entertainment consumption. By integrating two distinct participant groups, the end-users and professionals, this research aimed to capture both experiential and interpretive perspectives within a digitally transitional cultural context. Data were analyzed using a triangulated approach that included inductive and deductive coding, descriptive synthesis, and comparative matrix analysis. This multi-method framework enabled the researchers to extract thematic insights, compare divergences in perception between consumers and experts, and evaluate the emotional and cognitive layers shaping entertainment preferences in Romania.

### 3.1. Research Design

To explore the emotional, behavioral, and perceptual dimensions of entertainment consumption, qualitative research design was adopted. The first phase of data collection involved conducting focus groups with consumers aged between 18 and 27 who regularly engage with both physical and digital forms of entertainment. The method was selected for its capacity to capture in-depth attitudes, preferences, and lived experiences through interactive discussion ([13]). Participants were recruited using a screening questionnaire, and two age-based focus groups were formed: the first included individuals aged 18–22, and the second included participants aged 23–27. Each group consisted of eight participants, resulting in a total sample of sixteen respondents. Data collection took place between March and April 2024. Although data collection occurred in two phases (March–April 2024 for experts and January–February 2025 for consumers), all methodological tools and coding frameworks remained consistent across phases, ensuring comparability and minimizing temporal confounds. The segmentation was intended to reflect developmental differences in entertainment preferences and motivations across early and emerging adulthood. This research aimed to address five core objectives:(a)To explore consumer opinions on physical entertainment (festivals, concerts, tourism, parties, games);(b)To identify attitudes toward digital entertainment (streaming, music, gaming);(c)To uncover entertainment-related preferences and emotional needs;(d)To examine general behavioral patterns and decision-making processes;(e)To evaluate expectations regarding the future evolution of entertainment consumption.

While qualitative research does not aim for statistical generalizability, sample sizes were determined based on theoretical saturation principles. The two focus groups (16 participants) and seven expert interviews were sufficient to identify recurring themes without introducing redundant data. Participant selection was guided by pre-screening questionnaires that ensured strict alignment with selection criteria, as detailed in Section A.1 and Section A.2.

Each focus group session was guided by a semi-structured interview protocol featuring open-ended questions that encouraged participants to reflect on their personal experiences, motivations, and perceived barriers. Discussions were moderated to ensure thematic consistency across both sessions while still allowing participants to freely express views and elaborate on relevant issues. The responses were transcribed and subjected to content analysis using a mixed analytical strategy. Vertical and horizontal analyses were applied to identify recurring themes within and across focus groups. Descriptive and comparative techniques were employed to evaluate the frequency, depth, and variation in responses ([4]; [19]; [44]; [12]; [14]; [24]). Table 1 and Table 2 below present the demographic distribution of participants in each focus group. All participant information was anonymized, and informed consent was obtained prior to participation.

Participants were recruited from various urban and metropolitan areas across Romania, including Brașov, Bucharest, Cluj, and Iași. This approach aimed to capture a diversity of perspectives reflective of different regional contexts while recognizing the exploratory nature of qualitative research. This qualitative approach provided a foundation for interpreting consumer preferences in terms of stated choices and in relation to deeper emotional and psychological drivers. The insights generated in this phase were subsequently compared with expert perspectives in the second stage of this research, as detailed in the following paragraphs.

The second stage of qualitative marketing research was conducted through semi-structured interviews focused on entertainment industry ([46]; [50]). This phase involved the systematic questioning of highly qualified individuals with professional experience of at least two years in entertainment-related roles. The purpose of this research was to gather expert insights on consumer behavior, preferences, and needs in both physical and digital entertainment contexts. Data analysis was conducted using a combination of vertical and horizontal approaches, descriptive analysis, and comparative analysis ([12]; [14]; [24]).

The chosen method involved in-depth interviews structured around open-ended questions. This technique supports exploratory inquiry by encouraging detailed and reflective responses from participants ([35]; [41]; [51]). The interviewees were selected based on their direct involvement in the entertainment sector, including roles such as business owners, entrepreneurs, department heads, marketing directors, and marketing specialists. Proper identification of relevant experts ensured that the collected data would reflect well-informed and contextually grounded perspectives. The objectives pursued in the research phase included the following:(a)Discovering expert opinions regarding physical entertainment.(b)Identifying expert views related to digital entertainment.(c)Detecting the preferences and needs of the target audience as identified by experts.(d)Determining the behavior patterns of the target audience as interpreted by experts.

Seven professionals participated in this phase (see Table 3). Among them, three worked in physical entertainment fields, while four represented digital entertainment sector. Each interview lasted approximately 45 to 50 min. Data collection took place between January and February 2025. The methodological design aligns with the work of ([15]), who used in-depth interviews to examine how sales professionals employ metaphors to understand their customers. In this context, the experts’ discussions generated detailed insights into consumer behavior and motivations in the Romanian entertainment market.

Each interview was conducted using a structured guide that focused on four major thematic areas:A.Expert perspectives on physical entertainment.B.Expert perspectives on digital entertainment.C.Opinions about consumer preferences and needs.D.Interpretations of consumer behavior ([9]).

After the interviews were completed, data were analyzed using content analysis. The aim was to generate a rich understanding of expert insights by applying vertical and horizontal examination of themes, alongside descriptive and comparative techniques. The interview guide was developed using the open-question format, which aligns with qualitative research standards ([25]; [42]; [43]), and it facilitated the extraction of detailed perspectives on both physical and digital entertainment consumption patterns in Romania.

### 3.2. Inductive and Deductive Coding

The qualitative analysis in this study employed a dual strategy coding framework, integrating both inductive and deductive techniques to derive and interpret thematic insights from primary data source. This methodological approach enabled a comprehensive understanding of both experiential narratives and theoretically grounded interpretations. The inductive coding process was applied to the focus group transcripts and followed a bottom-up logic. Themes were allowed to emerge organically from participants’ spoken language and interactional patterns. Through a process of open coding and constant comparison, recurring concepts such as relaxation, social inclusion, and digital fatigue were identified and consolidated into broader interpretive categories. For instance, repeated references to festivals as “energizing” or “identity-affirming” were grouped under the emergent code of emotional anchoring. Latent constructs such as emotional safety, which were interpreted from implicit expressions of social anxiety or fear of judgment, were also extracted through inductive abstraction. This approach is consistent with established inductive coding methodologies, where themes are constructed directly from raw data without preconceived categorization ([9]).

In contrast, deductive coding was conducted on semi-structured interviews for expert interviewing process by using a top-down theoretical framework. This operationalization process of predefined categories such as motivation, barriers, digitalization, and identity construction was derived from the prior literature and conceptual models. Text segments were coded against these a priori themes to test and refine theoretical expectations. Notably, deductive analysis confirmed anticipated constructs such as the role of validation in entertainment choice and uncovered additional insights, such as the distinction between superficial inclusion and identity-guided mentorship. This aligns with contemporary practices in deductive qualitative analysis (DQA), wherein theory is iteratively evaluated, expanded, or refined through targeted coding ([22]).

To ensure analytic rigor and reliability, researchers independently coded the data, followed by intercoder comparison and theme consolidation through collaborative discussion. This iterative process reflects best practices in qualitative inquiry, where coding is both a technical and interpretive act aimed at reducing data complexity while preserving participant meaning ([38]).

To conduct this content analysis, the main themes were identified independently by two members of the research team. Some existing divergences were mediated by a third expert, with the aim of reaching a consensus. Validity of this study was assessed by using data triangulation ([16]). For each objective, the information obtained from the two categories of participants (consumers and industry specialists) was cross-checked and compared with the results of other research in the specialized literature.

### 3.3. Comparative Synthesis by Thematic Matrix

Following the coding phase, a comparative matrix analysis was conducted to analyze thematic representations from both consumer and expert data sources across seven analytical dimensions, namely, (i) motivational depth, (ii) barrier identified, (iii) emotional need recognition, (iv) clarity of entertainment preferences, (v) future consumption prediction, (vi) social connection insights, and (vii) identity construction awareness. These dimensions were selected based on their recurrence across inductive and deductive coding outcomes. To ensure objectivity, three researchers independently rated each theme on a 5-point Likert scale according to three criteria: (1) frequency of mentions, (2) interpretative depth (from superficial to multidimensional), and (3) diversity of sub-themes within the category. Thematic scores for both consumer and expert data were plotted on a radar chart, allowing a visual comparison of convergence and divergence in thematic emphasis. This visual representation facilitated a nuanced comparison of interpretive weight and cognitive framing between the two respondent groups ([8]). The radar chart was complemented by a comparison matrix, which offered a narrative representation of how each respondent group constructed meaning within each theme.

Ultimately, this triangulated synthesis, merging inductive insight generation, deductive theory validation, and comparative matrix analysis, provided a rigorous and multidimensional account of entertainment behaviors. This approach was particularly effective in illuminating the gap between behavioral patterns and the emotional architectures, offering valuable insights into culturally transitional contexts like Romania.

## 4. Results

The first part of this section presents the outcomes of primary data analysis. Within it, the authors examined perceptions of entertainment consumption in Romania through focus groups and semi-structured interviews with both consumers and specialists. After the primary data findings’ presentation, the researchers illustrated the insights of inductive and deductive categorization coding. In the final part, a comparison matrix was realized to cross-check, identify gaps, find trends over time, and support unconscious, latent needs.

### 4.1. Comparative Insights from Focus Groups and Semi-Structured Interviews

The findings show that Romania’s entertainment landscape offers a unique context marked by stark contrasts: a nation with one of Europe’s highest levels of digital connectivity yet an underdeveloped and fragmented physical entertainment infrastructure. This dichotomy makes Romania an ideal setting to study how individuals reconcile emotional and practical needs when choosing between physical and digital modes of entertainment. Specialists described the country’s entertainment options as being at a “starting level,” limited in diversity, visibility, and accessibility. One interviewee compared Romania with Hungary, noting, “Hungary didn’t have mountains, but they made their infrastructure friendly to tourism … Romania could exploit its geography more, but we see only viral trends like music festivals.”

The results of primary data analysis are organized into five thematic subsections that reflect the layered dynamics of emotional drivers, practical limitations, and evolving behaviors.

#### 4.1.1. Motivations and Barriers Related to Physical Entertainment

All sixteen consumers expressed a strong preference for physical entertainment. When asked to name their top three activities outside the home, festivals emerged as the dominant response, followed by socializing and sports. A typical response emphasized the emotional connection: “Festivals are not just fun; they help me feel connected. They’re where I find energy, where I feel seen.” These activities were valued for their capacity to generate motivation, happiness, and relaxation.

One respondent stated, “After going out, I feel more motivated at work and in my personal life.” Despite their enthusiasm, consumers acknowledged significant barriers to accessing physical entertainment. Six participants cited high costs as a deterrent, while others mentioned fatigue, lack of time, social anxiety, and fear of being judged. One consumer observed, “I’d love to go out more, but it’s expensive, and after work, I’m too tired. Sometimes I’m afraid of being judged too, it’s easier to just stay home.”

Experts offered a broader view on systemic barriers. They emphasized that Romania’s physical entertainment sector lacks public investment and is largely driven by private companies focused on profit. As one expert explained, “Only private companies offer entertainment, and they’re profit-driven. There’s no national visibility, no cohesive infrastructure.” The state was seen as having failed to harness Romania’s natural and cultural assets, leading to an underwhelming and uncoordinated physical offer.

#### 4.1.2. Accessibility and Emotional Shortcomings Related to Digital Environment

While digital entertainment was recognized as accessible, affordable, and convenient, its emotional value was repeatedly questioned. Consumers appreciated the practicality: “I can watch a movie or listen to music anytime without leaving the house or spending much. It fits my schedule.” However, 12 of the 16 consumers also expressed dissatisfaction with the quality of social interaction online. They described it as “low,” “shallow,” or “a loss of connection.” One respondent reflected, “Yes, I’m online all the time, but I don’t feel connected. It’s like being surrounded by noise but still feeling alone.”

Experts extended this critique, noting how digital platforms increasingly operate with commercial, rather than user-centered, priorities. “At first, digital was about giving access. Now it’s about exploiting users. They’re trapped in a loop that distances them from the real world,” one specialist stated. They further warned that sustained digital consumption may have cognitive repercussions: “Young people lose cognitive functions like attention, integration, reflection, because digital platforms demand less effort and more reaction.”

Despite these concerns, experts acknowledged Romania’s strengths in digital infrastructure. They noted the lack of restrictive policies on content, high internet speed, and a general cultural trend favoring online convenience, all of which contribute to digital entertainment’s dominant position.

#### 4.1.3. Emotional and Latent Needs

Participants’ stated desires, such as health, career, financial independence, and family, reflected common goal orientations. However, both consumers and experts highlighted deeper, often unarticulated, emotional drivers. One of the most consistent was the need for belonging, particularly among young people. Both consumer and expert data showed that entertainment was often a mechanism to fulfill this need, even if the consumers did not consciously realize it.

Consumers cited “socializing” as a top reason for preferring physical entertainment. Yet experts reinterpreted this as a proxy for more profound psychological needs. “They think they need to socialize, but really they seek validation. They want to be part of something that makes them feel real, to be seen, accepted, mentored,” one specialist said. Others highlighted how young people look to influencers and digital content for identity mentorship: “They take on behaviors from streamers not because they admire them, but because they fear being left out.”

Another latent need identified was guidance. Specialists observed that youth often feel overwhelmed by the abundance of options and information online. As one explained, “They have access to a lot, but they don’t know how to structure, prioritize, or choose. They need mentors, not more content.” These insights underscore a crucial point: while entertainment behaviors appear to be driven by convenience or fun, they are in fact embedded in deeper emotional ecosystems.

#### 4.1.4. Behavioral Contradictions

The data revealed consistent mismatches between consumers’ stated preferences and actual behaviors. Although every respondent declared a preference for physical entertainment, many admitted that digital entertainment dominated their routine when physical options were inaccessible. As one participant shared, “If I can’t go out, I scroll or stream something. It’s not the same, but it fills the gap, kind of.”

Experts interpreted this pattern as symptomatic of broader emotional and practical tensions. “They prefer physical, but it’s demanding socially, emotionally, even financially. Digital is easier, and once they’re in, it becomes a habit, even if it doesn’t satisfy them,” said one professional. This contradiction was reflected in emotional outcomes: while physical activities were linked with motivation, relaxation, and genuine joy, digital consumption was often described as distracting, temporary, or emotionally empty. “I’m happy after a festival, I feel alive. After a binge-watch, I just feel tired,” said a consumer, summarizing this dissonance. This behavioral gap suggests that the shift toward digital is not based solely on preference but also on emotional resignation and infrastructural realities.

#### 4.1.5. Future Preferences and Evolution

When asked about future trends, consumers were split. Nine believed their preferences would remain stable; seven foresaw change. Yet there was universal agreement that their needs would evolve, particularly as life priorities and time constraints shifted. As one respondent put it, “I don’t know if I’ll still love going to festivals in ten years, but I’ll still need to unwind. Maybe the how changes, not the why.”

Experts anticipated a more pronounced transformation. While they agreed that physical entertainment would remain desirable, they expressed concern that structural and behavioral inertia would make digital consumption increasingly dominant. “Digital will dominate, not because it’s better, but because it’s easier. That ease creates a cycle where people forget how to engage deeply, physically, emotionally,” one expert warned. They also noted that, without targeted interventions, such as infrastructure development or public cultural investment, physical entertainment might become a luxury, accessible only to a shrinking segment of the population.

More critically, they warned that entertainment is no longer merely a leisure pursuit; it has become a space where identity is constructed. “Entertainment is no longer just fun, it’s where young people build identity. If we let it become entirely digital, we risk creating a generation disconnected from reality,” concluded one specialist.

### 4.2. Inductive and Deductive Coding

The analysis of qualitative data involved both inductive and deductive coding approaches, applied, respectively, to the focus group and semi-structured interview responses. This dual coding framework allowed for the emergence of organic themes from consumers while also testing and refining expert-informed theoretical constructs.

The focus groups with consumers followed an inductive logic, where themes were derived directly from the participants’ language and interactions. Common categories emerged organically, including emotional anchors such as “relaxation,” “social inclusion,” and “motivation,” as well as barriers like “cost,” “time pressure,” and “digital fatigue.” For instance, repeated mentions of festivals as emotional recharge points clustered into a broader theme of identity-affirming entertainment. Similarly, consumer frustration around “being judged” or “feeling invisible” helped define a latent code: emotional safety as a prerequisite for physical participation.

In contrast, deductive coding was employed for expert interviews, where the semi-structured interviews guided directed participants through predefined themes: infrastructure, digitalization, motivations, barriers, and emotional interpretation. Here, the analysis confirmed expected theoretical dimensions (e.g., the role of validation and mentorship in behavioral choices) but also expanded them. For example, experts revealed a layered interpretation of belonging, differentiating between superficial socialization and deep-rooted needs for identity guidance. One noted, “They don’t just want to be included; they want someone to show them how to be.”

This mixed coding strategy enhanced the analytical rigor by enabling a bottom-up view from consumers and a top-down interpretation from specialists, facilitating later comparative synthesis. The key themes that appear in both the focus group (inductive) and expert interviews (deductive) are presented in Table 4.

For each category, the authors conducted a comparative assessment using a consistent 5-point Likert scoring rubric. Per group, researchers rated scores based on their overall subjective evaluation of frequency of relevant quotes or mentions, depth of insights (considering them surface-level or very nuanced), and range of sub-themes (in case of different types of motivations mentions). The scores (explained in the methodology of how they were rated) are shown in Table 5 and reflect that consumers (inductive coding) are more explicit about preferences and social needs, while experts (deductive) provide deeper emotional analysis and identity framing.

The focus group outcomes offered rich content on preferences and surface motivations, with several insights into emotional and social dynamics, whereas expert interviews revealed deeper layers of analysis, especially around identity, validation, and long-term societal impacts. To further synthesize the insights obtained, a radar chart (Figure 1) was constructed comparing thematic coding scores from focus groups (inductive analysis) and expert interviews (deductive analysis). The radar chart is intended as a visual, interpretive tool rather than a statistical model. The plotted scores reflect expert consensus ratings based on qualitative theme frequency and depth, not quantitative measures subject to statistical testing. This approach aligns with established qualitative visualization practices ([8]). The visual comparison across seven analytical categories illustrates both convergence and divergence in depth, focus, and interpretive richness between the two data sources.

The areas of greatest alignment include barriers to entertainment and social connection insights. Both consumers and experts emphasized a variety of obstacles to physical entertainment. Participants frequently cited issues such as cost, fatigue, and time scarcity, while experts extended these to include broader systemic and infrastructural deficits. In terms of social dimensions, both groups affirmed the value of entertainment in keeping human connection. Consumers focused on feelings of inclusion and group bonding, whereas experts interpreted these needs as symptomatic of deeper emotional drivers such as the pursuit of validation and psychological safety.

Moderate divergence was observed in the category of entertainment preference clarity. Focus group participants expressed highly defined preferences, overwhelmingly favoring physical entertainment forms such as festivals and live social experiences. Experts, while acknowledging these preferences, introduced greater nuance, noting a disconnection between what consumers claim to prefer and what they engage with, particularly in the face of digital convenience and inertia. A more pronounced discrepancy emerged in future consumption prediction. Consumers often assumed continuity in their behaviors or expressed uncertainty, while experts more consistently anticipated a trajectory of increased digital dominance, framed by concerns over emotional disengagement and structural inertia.

Substantial divergence occurred in three critical dimensions: motivational depth, emotional need recognition, and identity construction awareness. In terms of motivations, consumers articulated basic goals such as entertainment, relaxation, and social enjoyment. In contrast, experts interpreted these as proxies for deeper, often unconscious needs such as social validation, self-determination, and psychological resilience. The gap widened further in the domain of emotional needs. While participants occasionally mentioned emotional discomforts such as fear of judgment and feelings of isolation, their articulation of these remained surface-level. Experts, however, consistently identified latent needs such as guidance, belonging, and mentorship as central to entertainment engagement, particularly among youth. The most striking divergence appeared in the theme of identity construction. This topic was largely absent in consumer narratives but was prominently emphasized in expert interviews. Experts positioned entertainment as a central cultural mechanism through which young people develop, test, and adapt their identities, often unconsciously and with significant influence from digital platforms.

Considering all of these, the radar chart underscores a fundamental asymmetry in perspective: consumers reveal experiential, behavior-driven patterns, while experts provide interpretive frameworks grounded in psychological, emotional, and systemic analysis. The former emphasizes what is felt and done; the latter explains why it happens and what it signifies. This layered divergence reinforces the methodological value of combining inductive and deductive approaches and suggests that addressing emerging entertainment needs, especially in transitional markets like Romania, not only requires listening to consumer voices but also decoding the emotional structures that underlie them.

### 4.3. Comparison Matrix

While the radar chart visually emphasized interpretive asymmetries, the following comparison matrix (Table 6) enables a deeper, narrative-level juxtaposition. It enriches the radar findings by clarifying how much groups differ in insight depth and how they construct meaning differently around key themes.

This matrix confirmed that, although both groups share awareness of key trends such as rising digital consumption or the appeal of physical events, they differ sharply in depth of interpretation. Consumers focused more on immediate experience and circumstantial barriers, while specialists highlighted structural, psychological, and cultural dimensions. By bringing these two perspectives together, the matrix clarifies the current landscape and also identifies actionable gaps, particularly the unmet need for emotionally intelligent, identity-affirming entertainment environments.

These findings unveiled a crucial divergence between lived experience and professional interpretation, highlighting the need for more integrative strategies in understanding entertainment consumption. The implications of this divergence are further explored in the following Discussion Section.

## 5. Discussion

The present study aims to explore and compare the explicit and implicit entertainment needs and preferences of Romanian consumers, as interpreted by both end-users and industry specialists, to identify emerging behavioral patterns, emotional drivers, and strategic implications for the entertainment sector within a digitally advanced but infrastructurally evolving market context.

Findings highlight that both consumers and specialists recognize entertainment as a space for social connection and temporary relief, yet their interpretations diverge in depth. Consumers describe motivations in terms of enjoyment, relaxation, and inclusion. Experts, on the other hand, interpret these behaviors as proxies for latent needs such as validation, identity formation, mentorship, and emotional safety. These underlying drivers remain largely unspoken by consumers but critically shape their entertainment choices. Barriers to physical entertainment, such as cost, fatigue, and social discomfort, are compounded by Romania’s fragmented infrastructure and lack of public investment in cultural offerings. Experts emphasize that the absence of cohesive, accessible physical entertainment spaces exacerbates reliance on digital options, reinforcing emotional resignation rather than conscious preference.

Digital entertainment, though pervasive, is widely regarded by both groups as emotionally shallow. Consumers describe it as convenient but ultimately isolating, while experts raise concerns about its long-term cognitive effects and its role in promoting superficial social engagement. Despite this, digital formats are likely to continue dominating due to behavioral inertia and the lack of viable physical alternatives. [39] ([39]) propose a framework categorizing these impacts into four scenarios: disastrous, silver lining, fortuitous, and triumphant, based on the direction (positive or negative) and continuity (temporary or lasting) of the changes. This analysis underscores the need for the entertainment industry to embrace digital transformation to enhance resilience against future disruptions.

Ultimately, this study exposes a layered asymmetry between experiential accounts and expert interpretations. It highlights the need for emotionally intelligent entertainment strategies that address both articulated preferences and unspoken psychological needs. Bridging this gap is essential for designing culturally responsive, identity-affirming entertainment experiences in digitally saturated yet infrastructurally evolving societies like Romania. Based on the identified insights, the authors propose the following actionable strategies: develop modular event formats with flexible pricing structures to reduce cost barriers; integrate mentorship components into entertainment offerings, such as guided experiences or influencer-led community events; enhance digital platform features by embedding tools that foster social connection and identity affirmation, such as customizable user profiles and moderated community spaces; advocate for public investment in cultural infrastructure to diversify physical entertainment access points; and introduce hybrid entertainment models blending digital convenience with physical emotional depth, such as virtual festivals with local meetup hubs.

Moreover, this section interprets the findings in alignment with the research questions relevant to entertainment marketing strategy presented in the Introduction. The Discussion is structured along these three axes and emphasizes interpretive depth, theoretical relevance, and practical implications sustained by outcomes (Figure 2).

Specialists in the field offered a comprehensive and reflective understanding of entertainment consumption, emphasizing structural patterns and emotional frameworks. Their interpretation of the Romanian market suggests a high potential for growth through the strategic development of physical entertainment experiences. Experts consistently described entertainment as a leisure activity and as a platform for psychological anchoring. They highlighted the relevance of mentorship, validation, and identity formation as underlying needs frequently expressed in subtle forms by younger audiences. From a marketing standpoint, these insights offer valuable opportunities. Brands that align their messaging and service design with themes of personal development, emotional connection, and social recognition are better positioned to engage youth segments. Specialists also observed that entertainment functions as a cultural medium where identity is shaped, which indicates the importance of emotionally intelligent content in long-term brand strategies.

Consumers expressed clear preferences and emotional affiliations with physical entertainment. Their frequent references to festivals, social gatherings, and active leisure reveal a desire for immersive and energizing experiences. These preferences suggest that entertainment brands can increase impact by focusing on emotional resonance, personal relevance, and collective experiences that evoke belonging and self-expression. Consumers also described specific limitations influencing their entertainment choices, including financial accessibility, time management, emotional fatigue, and social discomfort. These constraints present valuable touchpoints for marketers. Strategic pricing models, modular and scalable event formats, and inclusive communications can transform barriers into opportunities for participation. By enhancing comfort and reducing social risk, brands can expand their appeal to hesitant or underserved segments.

Moreover, consumers frequently engage with digital entertainment due to its accessibility and integration into daily routines. This behavioral tendency reveals an important insight for content designers and experience marketers. While digital environments may not currently elicit strong emotional fulfillment, they provide a reliable channel to introduce and promote more engaging alternatives.

The radar chart highlighted shared insights, such as the centrality of social connection and the existence of structural barriers, as well as disparities, particularly in dimensions like identity construction and motivational depth, where expert interpretations were significantly more layered. While consumers frequently described entertainment as a form of inclusion and joy, experts framed it as a psychologically loaded space for identity negotiation and emotional validation. This dual reading underscored the asymmetry in how lived experience and expert interpretation articulate the same phenomena.

The combination of radar analysis and matrix comparison enables a dual-layered interpretation of consumer behavior. Consumers and specialists both recognized key drivers such as social inclusion and emotional connection. However, the two groups approached these drivers from different cognitive and emotional depths. Consumers spoke primarily through the lens of experience and preference. Specialists provided interpretations rooted in emotional frameworks, behavioral theory, and structural observations.

This difference enhances marketing insight. It points to the importance of designing entertainment offerings that reflect both felt experiences and unspoken aspirations. For instance, while many consumers prioritize enjoyment and social bonding, their deeper emotional needs include personal recognition, identity exploration, and structured guidance. These needs create space for entertainment brands to become lifestyle partners, not only experience providers.

The comparative framework also reveals patterns that assist in segmenting audiences and customizing brand engagement. Consumers’ emphasis on convenience and mood alignment highlights the value of omnichannel offerings and emotionally framed content. Specialists’ emphasis on latent motivations and identity development supports the creation of value propositions that inspire long-term involvement and psychological loyalty.

The findings affirm the growing relevance of emotionally intelligent branding in entertainment marketing. Entertainment no longer serves a purely recreational function. It now shapes lifestyle choices, emotional states, and identity construction. In this context, brand strategies that prioritize value-based communication, adaptive experience design, and psychological resonance will likely yield stronger consumer relationships.

In emerging markets such as Romania, the coexistence of high digital readiness and underdeveloped physical infrastructure creates unique opportunities. Brands that offer community-centered, emotionally affirming, and structurally accessible experiences can fulfill a dual role: satisfying immediate preferences and supporting deeper developmental needs. This approach encourages innovation in product design, promotional storytelling, and audience engagement.

By synthesizing the diverse perspectives gathered through inductive and deductive approaches, marketers gain a richer understanding of their audiences. Consumers’ explicit expressions reveal behavioral preferences, while specialists’ interpretations provide a foundation for strategic foresight. This integrated perspective enhances the ability of entertainment providers to design offerings that are enjoyable and also meaningful and culturally relevant.

## 6. Conclusions

This research contributes to the existing body of knowledge by exploring and de-fining the key traits of consumption behaviors in the entertainment industry, providing a theoretical framework for future research and practical applications. This paper highlights consumption behaviors that are guided by very specific desires and needs that reach a deep emotional level, such as validation, identity formation, psychological security, and belonging. In this sense, both analyses highlight how experts and consumers view entertainment consumption. Thus, consumers have a very clear understanding of their consumption preferences. However, experts, approaching the analysis from an external and objective standpoint, are able to identify a broader range of factors that shape consumer identity, including motivations, barriers, and social, emotional, and psychological stimuli.

These dynamics arise because consumers tend to focus on the outcomes of their actions and the immediate effects they experience. In contrast, experts, drawing from an external and objective perspective, are better positioned to identify the underlying causes of consumer behavior and to interpret how these causes influence individual experiences and identity formation.

Although consumers show a strong preference for physical entertainment and experts recognize its more substantial impact, both groups identify significant barriers, such as accessibility, cost, and time, that largely account for the increased reliance on digital entertainment as a functional substitute.

This difference reveals a contrast in depth of understanding: consumers tend to view digital entertainment as a fallback when physical options are unavailable, while experts recognize digital entertainment as the primary substitute, routinely adopted in the face of structural limitations.

In analyzing these findings, the motivations of contemporary consumers can be interpreted through both Maslow’s hierarchy of needs and the ERG theory. While Maslow suggests that basic needs must be met in sequence from physiological to self-actualization, ERG theory proposes that these needs can exist simultaneously and be pursued in parallel. This study underscores the prominence of belonging as a core consumer motivation. In today’s socially driven context, the desire to belong is often amplified by social pressure, and behaviors such as socialization can evolve from intrinsic motivation to perceived obligation. This shift illustrates how foundational components of Maslow’s hierarchy, such as the need for social connection, may assume new, context-dependent roles.

This study acknowledges several limitations. First, the qualitative data cannot be generalized to the broader population. Nevertheless, the insights derived are highly relevant to the research topic and provide a strong foundation for future marketing strategies. Second, not all market experts were deemed representative or available for participation. Third, some experts may have misunderstood certain questions. However, this limitation was addressed by the researchers through clarification during interviews. Fourth, although participants were initially identified through networks associated with industry experts, actual recruitment was conducted independently using pre-screening questionnaires. Therefore, the focus groups did not consist of the experts’ own consumer base, ensuring that consumer perspectives were gathered without direct influence from the participating specialists. This adjustment mitigates potential bias and strengthens the reliability of identity-related insights. Another limitation of this study is that initial demographic reporting was incomplete regarding participants’ socioeconomic status, educational background, and detailed entertainment consumption patterns. Future research could benefit from integrating these variables more systematically into both data collection and analysis. The findings also reveal that many younger consumers are unaware of the depth of their emotional needs. Experts observed that youth often express a strong desire for belonging, whether through a group, community, or personal relationships, which stems from a deeper need for validation. Experts operating in physical environments are better positioned to observe these dynamics directly and recognize how the need for belonging is often driven by an underlying need for reassurance. In an era of overwhelming choices and information, many young people seek guidance and affirmation, indicating a possible decline in self-confidence and an increased reliance on mentorship for decision making.

In contrast, the digital environment offers a more controlled and private space, where individuals can curate their identities and manage interactions with less exposure. This creates a different dynamic, wherein social engagement occurs on the consumer’s terms, and anonymity may offer a sense of security. However, this does not necessarily translate into emotional fulfillment.

Based on these insights, several practical recommendations emerge. For professionals in physical entertainment, it is essential to recognize and address consumer insecurities by fostering environments that feel emotionally safe and supportive. Those responsible for direct customer engagement should adopt a mentorship-oriented approach, helping consumers navigate experiences in ways that enhance satisfaction and perceived belonging.

For professionals operating in digital environments, the emphasis should be placed on the platform’s unique strengths, especially its capacity to offer a safe, controlled space for communication. By promoting social interaction and reinforcing community, digital services can continue to satisfy consumers’ needs for belonging and identity affirmation, even in the absence of deep emotional stimulation.

The findings from this study offer a valuable basis for continued research. Future directions should include quantitative investigations into consumer behavior within the entertainment industry to validate and expand upon the qualitative insights presented. Additionally, further research should explore consumer reactions to decisions made by experts in the field, allowing for a more complete understanding of how professional practices influence user experience and preference formation.

## Figures and Tables

**Figure 1 behavsci-15-01049-f001:**
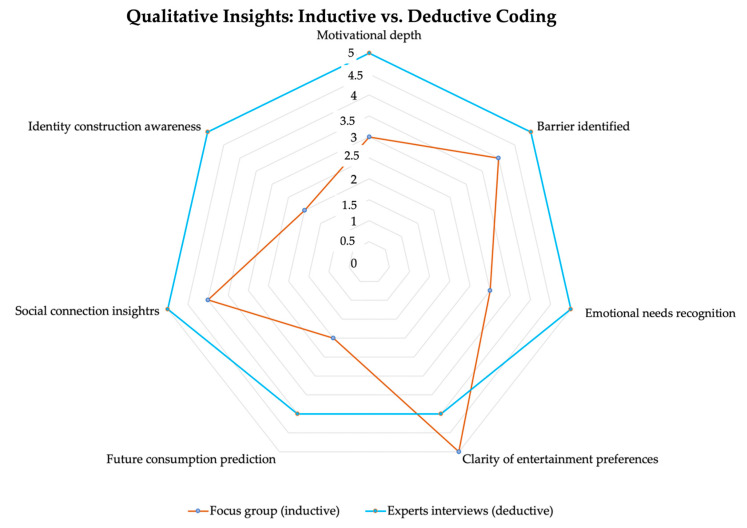
Radar chart: Qualitative insights: inductive and deductive coding. Source: Created by authors based on respondents’ insights.

**Figure 2 behavsci-15-01049-f002:**
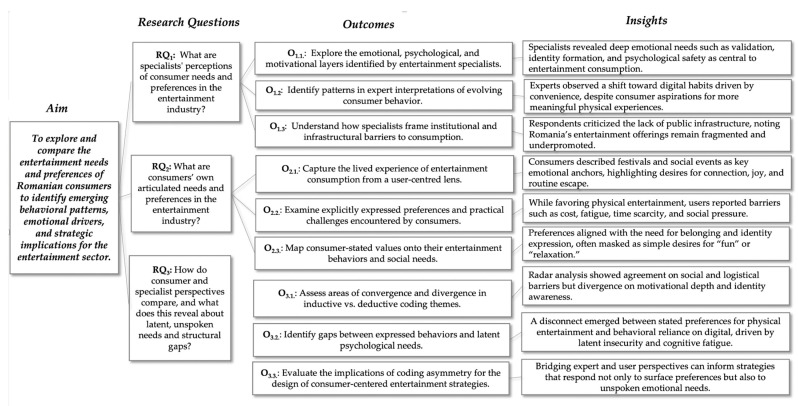
Research outcomes based on aim and research questions. Source: Created by authors.

**Table 1 behavsci-15-01049-t001:** The demographic characteristics of the first focus group.

Code	Genre	Age
S1	Feminine	18–22 years
S2	Masculine	18–22 years
S3	Masculine	18–22 years
S4	Feminine	18–22 years
S5	Feminine	18–22 years
S6	Masculine	18–22 years
S7	Feminine	18–22 years
S8	Feminine	18–22 years

Source: Created by authors.

**Table 2 behavsci-15-01049-t002:** The demographic characteristics of the second focus group.

Code	Genre	Age
S1	Feminine	23–27 years
S2	Feminine	23–27 years
S3	Feminine	23–27 years
S4	Masculine	23–27 years
S5	Masculine	23–27 years
S6	Masculine	23–27 years
S7	Feminine	23–27 years
S8	Masculine	23–27 years

Source: Created by authors.

**Table 3 behavsci-15-01049-t003:** Demographic characteristics of the interviewed experts.

Code	Genre	Age	Occupation	Experience
S1	Feminine	30–39 years	Dance Studio Director	2 years
S2	Masculine	40–49 years	Music Studio Director	21 years
S3	Masculine	40–49 years	Travel Agency Director	11 years
S4	Masculine	30–39 years	Entrepreneur	4 years
S5	Masculine	40–49 years	Marketing Director	4 years
S6	Masculine	20–29 years	Marketing Specialist	3 years
S7	Masculine	30–39 years	Marketing Specialist	3 years

Source: Created by authors. All participants provided consent for anonymized inclusion of demographic data.

**Table 4 behavsci-15-01049-t004:** Shared categories and coding dimensions.

Category and Coding Dimension	Description
Motivational depth	How deeply respondents articulated the reasons behind entertainment engagement, ranging from basic enjoyment to underlying psychological or social drivers like identity affirmation or personal growth.
Barrier identified	The range and complexity of factors that prevent or limit participation in entertainment activities, including logistical, emotional, financial, or systemic obstacles.
Emotional need recognition	The extent to which participants, explicitly or implicitly, express emotional motivations such as belonging, validation, or psychological safety as underlying drivers of entertainment behavior.
Clarity of entertainment preferences	How clearly and consistently respondents were able to define their favored types or formats of entertainment (festivals, digital content), and the reasoning behind those preferences.
Future consumption prediction	How participants perceive the trajectory of their entertainment habits (whether as stable, evolving, or subject to external forces like technology, social change, or infrastructure development).
Social connection insights	The extent to which respondents perceive entertainment as a mechanism for building, maintaining, or avoiding social interactions and relationships.
Identity construction awareness	Measures the awareness participants show regarding the role of entertainment in shaping personal or collective identity, including adoption of values, behaviors, or group norms.

Source: Created by authors based on respondents’ insights.

**Table 5 behavsci-15-01049-t005:** Theme scores.

Category	Focus Group (Inductive)	Experts Interviews (Deductive)
Motivational depth	3	5
Barrier identified	4	5
Emotional need recognition	3	5
Clarity of entertainment preferences	5	4
Future consumption prediction	2	4
Social connection insights	4	5
Identity construction awareness	2	5

The authors applied Likert scale: from 1—superficial, 3—moderate elaboration, some nuance, to 5—frequent, even multidimensional discussion.

**Table 6 behavsci-15-01049-t006:** Comparison matrix.

Theme	Consumers	Specialists
Motivational depth	Happiness, belonging, self-expression	Validation, mentorship, social identity construction
Barrier identified	Cost, time, social anxiety	Structural lack, fragmented offer, low institutional support
Emotional need recognition	Recognition, escape from routine, group acceptance	Deep need for guidance, role models, and structured identity formation
Clarity of entertainment preferences	Strong preference for physical entertainment (festivals, social events)	Acknowledge consumer preference but note digital dominance due to convenience
Future consumption prediction	Mixed, some anticipate changes, others hope for stability	Predict dominance of digital due to inertia; express concern over detachment
Social connection insights	Entertainment as inclusion, bonding, temporary motivation	Indicator of deeper validation and identity needs
Identity construction awareness	Largely absent or implicit	Physical offers longer-lasting impact; digital creates short-term reward loops

Source: Created by authors based on respondents’ insights.

## Data Availability

Data is contained within the article.

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
