# Peer review of "Uncovering Emotional and Identity-Driven Dimensions of Entertainment Consumption in a Transitional Digital Culture"

_behavsci, 2025, doi:10.3390/bs15081049_

Round 1
Reviewer 1 Report
Comments and Suggestions for Authors
1.Sample Size and Generalizability: The study employs very small samples (16 focus group participants, 7 experts) without adequate justification for these sample sizes or discussion of saturation. The authors acknowledge generalizability limitations but fail to provide sufficient theoretical justification for their approach.
2.Bias and Validity Concerns: A critical flaw is revealed on page 18 where the authors note that "focus groups consisted of the experts' own consumer base." This creates substantial bias as consumers may have been influenced by their relationship with the service providers, potentially skewing responses toward more favorable views of the experts' businesses.
3.Geographic Limitations: The study claims to examine "Romania" broadly but provides no information about geographic distribution of participants within the country, potentially limiting representativeness even within the stated scope.
4.Radar Chart Interpretation: The visual comparison tool, while aesthetically appealing, lacks statistical rigor and may overemphasize differences that could be within measurement error.
5.Limited Theoretical Advancement: The study primarily confirms existing knowledge about digital vs. physical entertainment preferences without offering substantial new theoretical insights. The connection to "transitional digital culture" remains underdeveloped.
6.Superficial Literature Review: The literature review lacks depth in connecting entertainment consumption to broader theories of consumer behavior, identity formation, and cultural transition. Key theoretical frameworks are mentioned but not rigorously applied.
7.Timeline Confusion: Data collection occurred at different times (March-April 2024 for consumers, January-February 2025 for experts), potentially introducing temporal confounds.
8.Inconsistent Data Reporting: Demographic information is incomplete (e.g., Tables 1-3 lack important contextual details about participants' backgrounds, socioeconomic status, or entertainment consumption patterns).
Author Response
Please, see the attachment!

Reviewer 2 Report
Comments and Suggestions for Authors
This paper investigates the emotional and identity-driven dimensions of entertainment consumption in Romania, within the context of a transitional digital culture. The topic is timely and relevant, and the manuscript is well-organized. However, the following questions remain:
1. The description of participant recruitment and interview procedures is rather brief. Provide details on recruitment channels, how the focus‐group protocol and interview guide were pilot-tested, and what revisions were made based on that piloting.
2. Although themes such as “digital fatigue” and “cognitive overload” are mentioned, there is insufficient reference to localized studies or recent quantitative research. Strengthen the argument by incorporating regional studies (e.g., Eastern or Central European contexts) or recent empirical quantitative work on these phenomena.
3. The coding procedure is described professionally, but important quality-control steps—such as homogeneity checks and reliability metrics—are not specified. Report inter-coder reliability statistics (e.g., Cohen’s κ), describe any triangulation procedures, and explain how coder agreement was established and monitored.
4. While the results are rich, the numerical values in figures (radar charts, comparison matrices) and their statistical interpretations lack depth. Include means and standard deviations for each dimension and conduct significance tests (e.g., Wilcoxon signed-rank or paired t-tests) to compare groups more rigorously.
5. The practical suggestions in the Discussion and Conclusion are somewhat general and lack actionable detail. Based on interview insights, propose three to five measurable, implementable items—such as specific event-design elements, digital-platform feature enhancements, or policy-advocacy steps—to guide practitioners.
Author Response
Please, see the attachment!

Round 2
Reviewer 2 Report
Comments and Suggestions for Authors
The authors did a great job answering all of my questions about the paper.